# An Optimal Footprint Based Coverage Planning for Hydro Blasting Robots

**DOI:** 10.3390/s21041194

**Published:** 2021-02-08

**Authors:** Thejus Pathmakumar, Madan Mohan Rayguru, Sriharsha Ghanta, Manivannan Kalimuthu, Mohan Rajesh Elara

**Affiliations:** Engineering Product Development Pillar, Singapore University of Technology and Design, Singapore 487372, Singapore; pathmakumar_thejus@mymail.sutd.edu.sg (T.P.); madan_rayguru@sutd.edu.sg (M.M.R.); ghanta_sriharsha@mymail.sutd.edu.sg (S.G.); manivannan_kalimuthu@mymail.sutd.edu.sg (M.K.)

**Keywords:** coverage planning, hydro blasting robot, multi-objective optimization, optimal footprint

## Abstract

The hydro blasting of metallic surfaces is an essential maintenance task in various industrial sites. Its requirement of a considerable labour force and time, calls for automating the hydro blasting jobs through mobile robots. A hydro blasting robot should be able to cover the required area for a successful implementation. If a conventional robot footprint is chosen, the blasting may become inefficient, even though the concerned area is completely covered. In this work, the blasting arm’s sweeping angle is chosen as the robot’s footprint for hydro blasting task, and a multi-objective optimization-based framework is proposed to compute the optimal sweeping arc. The genetic algorithm (GA) methodology is exploited to compute the optimal footprint, which minimizes the blasting time and energy simultaneously. Multiple numerical simulations are performed to show the effectiveness of the proposed approach. Moreover, the strategy is successfully implemented on our hydro blasting robot named Hornbill, and the efficacy of the proposed approach is validated through experimental trials.

## 1. Introduction

In recent times, the ship hull maintenance has become a prominent area of interest in robotics. Ship hull maintenance is one of the key marine sector activities that thrive on automation technologies. The major tasks involved in ship hull maintenance include hull cleaning, hull paint removal, re-painting and welding. Cleaning of a ship hull that is infested with tightly sticking barnacles (Biofouling) requires high-pressure washing [1,2]. In the case of paint removal, an extremely high-pressure hydro-blasting or abrasive blasting is inevitable [3,4]. Therefore, ship hull cleaning and hull biofouling management using hydro-blasting are often considered unsafe and time consuming task in the marine industry. Automated robotic solutions for ship-hull hydro-blasting are inevitable for productivity enhancement and ensuring a safe working environment for human workers. For the past decade, we can observe evidence for a significant research effort to develop robotic solution for ship-hull hydro-blasting. A remotely operable anti-fouling robot, called ARMROV is proposed in [5], where the authors introduced a under water robot with two manipulators that perform the cleaning task. Morgan. H proposed a novel mechanism to increase efficiency by minimizing lateral force resistance on ship-hull cleaning robot [6]. Zheng et al. proposed a semi-automatic robotic solution to ease the manual ship-hull cleaning [7]. Milella et al. proposed ship-hull inspection method using robot that is equipped with magnetic tracks which aids the robot in climbing and a monocular camera for the inspection purposes [8].

Autonomy is an integral part of any robot, and considerable research work has been reported for novel autonomy strategies in ship-hull cleaning robots, especially in localization, perception and control strategies. For instance, the research mentioned in [9], discusses a method for localization on an autonomous ship-hull cleaning robot using optical displacement sensors. The research in [10] proposes a method to enhance the localization based on wheel-drift correction. Muthugala et al. proposed a novel fuzzy-logic based strategy for classifying and bench-marking hull hydro blasting on an autonomous ship cleaning robot [11].

Besides the perception and localization, new methods and strategies for field coverage algorithms have been widely researched. In [12], authors proposed an area coverage algorithm that is modular, cost-effective and computationally efficient for heterogeneous mobile robots. Yongbo et al. [13], proposed an active SLAM framework for collision-free trajectory and area coverage for mobile robots using predictive control model. The authors in [14] have demonstrated a Robot Operating System-based multi-node control of a small-scale mobile robot. Coverage Path Planning recently has gained much focus in robotics, which aids in efficient area coverage. Phone et al. in [15] implemented a framework for optimizing the coverage path planning using deep reinforcement learning based on the travelling salesman technique. Authors in [16] have shown an efficient method for area coverage of a known area using multiple mobile robots. They have applied cellular decomposition method and the greedy search approach to achieve complete area coverage. Cell permeability-based coverage (CPC), is a new coverage path planning algorithm proposed by [17]. Le et al. proposed complete coverage path planning on Tetris-inspired re-configurable robot [18]. Even though a vast volume of research work is available on complete coverage path planning, only rudimentary focused is given on solving the challenges in complete coverage path planning in surface propagation robots. Le et al. proposed a coverage path planning strategy for hydro blasting using Deep Learning and Evolutionary algorithm [19]. Similarly, a path planning strategy using an evolutionary algorithm for underwater cleaning robot is discussed in [20].

The robot footprint is an essential aspect of area coverage algorithms, which conventionally defined as the surface area occupied by a robot at any instants of time [21]. For a complete coverage problem, the robot should move so that its footprint covers the whole area. Similar concepts are present full sensory coverage, or a visual coverage [22]. For completing a task like painting or hydro blasting, the task completion depends on the robot’s ability to reach every point and how the task is performed at each step. If the robot footprint is chosen based on the shape of the robot, the task-driven autonomy may suffer. In that sense, a different footprint definition based on the robot’s function (blasting, painting) (called functional footprint in this paper) can be useful. For example, the painting area can act as a functional footprint for painting automation jobs at any instantaneous time interval (area covered by the robot when it is not propagating).

Even if the footprint is chosen according to a particular task, it may not be optimal. For any autonomous robot, the task completion time and energy consumed while completing a task have a direct impact on the usability of the robot. Some heuristic logic like “minimum path will spend minimum time and energy” may not work for hydro blasting because a complete coverage path may not be the shortest path. Moreover, the time taken for sweep/blast and energy consumed during the process plays a significant role, apart from the platform motion. Once the coverage strategy is decided, the blasting task directly impacts the time and energy consumed during the whole process. So far, the area coverage algorithms for hydro blasting have not explicitly considered the time and energy consumed by the robot during blasting.

This paper aims to bridge this gap in the area coverage planning for hydro blasting robots. In this direction, the major contributions of the paper are:Two separate coverage strategies called line sweep and stop sweep are proposed for complete coverage in Hornbill hydro blasting robots. The surface area covered by the nozzles in one complete sweep is considered as the robot’s functional footprint.Unlike the earlier methods, explicit expressions of time and energy spent during blasting are obtained. The expressions are derived in terms of sweeping angle and speed of the nozzle arm. These expressions are considered as cost functions of a multi-objective optimization problem.The optimal footprint (sweeping angle) is computed through a multi-objective GA, and the obtained results are implemented on the Hornbill robot. It is found that the Hornbill robot can minimize its time and energy during blasting apart from area coverage.

The paper is organized in the following way—Section 2 describes two types of Hornbill robots for automating the hydro blasting task. The sweeping strategies required for complete coverage is presented in Section 3. Section 4 derives the cost functions related to blasting time and energy in terms of the sweeping angle, followed by the Section 5. Finally, concluding remarks are given in Section 6.

## 2. Hornbill Architecture

The detailed mechanical structure and electronics layout is discussed in [23]. We briefly describe the architecture in this section for the completeness of the paper. The Vertical surface propagation robot, Hornbill comprises of a locomotion module with independent powered wheel, a magnetic module to establish adhesion to the ship hull, a load carrying swivel caster wheel. Furthermore, the robot is equipped with a high-pressure cleaning unit to carry out the painting, barnacle removal and paint removal. The robot’s traction wheel is powered using two 48 V, 400 W Brush-less DC (BLDC) motor coupled with a 100:1 epicyclic gear unit. The isometric view of the open arm version and enclosed arm version of Hornbills are shown in Figure 1. To prevent the free rotation of the wheels, the BLDC motors are equipped with electromagnetic brake which can be activated using electrical signals.

As shown in Figure 2 and Figure 3 the adhesion unit incorporates fifteen N52 grade Neodymium rare earth magnets, each has a length, width, height of 100 mm, 25 mm, and 25 mm, respectively. To provide corrosion resistance each magnet treated with 10 micron three-layer coating of Ni-Cu-Ni. And single magnet exerts 138.25 kgf in the orthogonal direction when the air gap is zero.

In order to increase the robot’s traction with the smooth and wet vertical metal surface, wheels are customized to meet the 80A shore hardness rubber tread which is produced using the low volume casting method. As shown in Figure 4.

### System Architecture

As shown in Figure 5, serial communication is utilized for communicating between robot and base station: power source, communication cable and water supply for the robot provided from the ground. The operator can control the robot’s linear speed, direction, desired arm velocity, and the number of movement cycles per area. The robot can be controlled using a wireless joystick or by the software node running inside the Embedded industrial PC (IPC). To simplify the overall system camera, Inertial Measurement Unit (IMU) sensor and motor controllers are plugged to IPC through a single USB. MODBUS [24] protocol in RTU transmission via RS485 interface utilized to send/receive/modify information between IPC (Master) and Robot (Slave) addresses. Shielded cables are employed to prevent the signal loss over a long transmission line of 30 m. Embedded Industrial PC that powers the robot runs on Ubuntu 16.04 LTS operating system (OS). The top-level navigation, control algorithms, necessary software nodes for hardware integration are hosted on the IPC.

## 3. Coverage Strategies

The hydro blasting robot is supposed to cover the whole area under consideration. This section presents two different coverage strategies for the Open-Arm Hornbill robot mentioned in the previous section.

### 3.1. Line-Sweep Strategy

For the fixed arm Hornbill robot, the blasting nozzles are attached to two fixed arms. In such a case, a lawn-mower type area coverage is suitable, which is termed as line-sweep (LS) strategy (c.f. Figure 6).

In this strategy, the functional footprint ($f_{r}$) is considered as the maximum circular area, which the blasting nozzles can cover when the robot is stationary. Given a rectangular area cross-section (L×W,L>W), the whole surface is divided into $n_{s}$ number of strips, where
(1)$$n_{s}=int\left(\frac{W}{f_{r}}\right)+1.$$

*W* is the width of the area to be covered by the robot, $f_{r}$ is the functional footprint. The function “*int*( )” rounds off its argument to the largest integer which does not exceed the argument and the expression of $n_{s}$ is guaranteed to be an integer. Once the area is divided into a fixed number of strips, the robot moves along each strip until the length (*L*) is covered. At the end of each strip, the robot rotates $180^{\circ }$ degrees and moves a distance of $f_{r}$ to reach the next strip. This is repeated till the robot reaches the end of the last strip.

### 3.2. Stop-Sweep Strategy

For the moving arm Hornbill robot, the whole area can also be covered through LS strategy. However, the extra degree of freedom of the moving nozzle arm will be of no use in that case.

A simple but effective coverage strategy will be to stop at fixed lengths (*B*); sweep the nozzle through a predefined angle ($-\frac{\alpha }{2},+\frac{\alpha }{2}$); and repeat the process till the complete length *L* is covered (c.f. Figure 7). The SS method’s significant difference is the inclusion of robot’s functional footprint, which is now dependent on the sweeping arc. The number of strips $n_{s}$ depends on the angle α (sweep angle). For a larger α, the $n_{s}$ can be small, but the arcs for each sweep may overlap. This overlap of the sweeping arcs can lead to wastage of more energy and time. If the sweeping arc is kept under a specific range (depending on the nozzle arm length and robot structure), the overlap can be avoided. For the concerned Hornbill robot, it is found that a sweep angle in excess of $60^{\circ }$ range leads to arc overlap, and therefore the sweep angle has to be within this range for our setup.

## 4. Optimal Coverage Planning

The time taken for an LS strategy depends on the robot velocities. Similarly, the time taken for the blasting and the required energy demand varies with different sweeping angles, even though the sweeping angle (α) is kept within $60^{\circ }$. Note that a sweeping angle for minimum time may not be optimal in terms of energy consumed, and vice versa. Therefore, time and energy costs must be separately considered for choosing an optimal footprint (sweeping angle). This section focuses on deriving the necessary cost functions for minimum time behaviour for both the blasting strategies.

For both the strategies, the total time required for the blasting can be derived as:(2)$$T_{blast}=T_{sweep}+T_{translation}+T_{rotation},$$
where the subscripts denote the operation robot is performing.

### 4.1. Line-Sweep

In this strategy, the robot moves without the nozzle rotations. The number of strips required for complete area coverage can be derived as:(3)$$n_{s}=\frac{W}{d_{n}},$$
where $d_{n}$ is the diameter of the nozzle.

The sweep time is equal to the time required for translations. Hence,
(4)$$T_{sweep}=T_{translation}=n_{s}\frac{L}{v_{l}},$$
where $n_{s}$ is the number of strips along the total area, $v_{l}$ is the linear velocity of the robot.

**Note:** Even though the nozzle diameter can be taken as a possible choice for $d_{n}$, proper care should be taken such that $n_{s}$ is an integer. A straightforward alternative is to round down the nozzle diameter to get an integer value of $n_{s}$. However, it may lead to more strips to move around. Another alternative is to choose the highest integer which is below $\frac{W}{d_{n}}$ as the number of strips, plus an additional strip of smaller width than $d_{n}$.

The time required for the translation and sweep through the complete area can be written as:(5)$$T_{sweep}=T_{translation}=\frac{LW}{d_{n}v_{l}}=\frac{A_{blast}}{d_{n}v_{l}},$$
where $A_{blast}$ is the area of rectangular cross-section selected for blasting. Apart from these movements, the robot needs to move from one strip to next after completing the blasting along each strip. For moving into next strip, the robot needs to rotate 90 degrees (π/2 radian); move $d_{n}$ distance straight and rotate 90 degrees (π/2 radian) again. For the total area, this operation has to be repeated $\left(n_{s}-1\right)$ times. So,
(6)$$\begin{array}{c} T_{rotation}=\left(n_{s}-1\right)\left(\frac{\pi }{\omega_{l}}+\frac{d_{n}}{v_{l}}\right),\\ T_{blast}=\left(n_{s}-1\right)\left(\frac{\pi }{\omega_{l}}+\frac{d_{n}}{v_{l}}\right)+\frac{A_{blast}}{d_{n}v_{l}}, \end{array}$$
where $\omega_{l}$ is the angular velocity of the platform.

### 4.2. Stop-Sweep

For this strategy, the time taken for translation ($T_{translation}$) remains the same. However, the computation of $T_{rotation}$ and $T_{sweep}$ changes due to the change in the width of each strip ($S_{w}$). It can be observed from Figure 7 that, the strip width $S_{w}$ depends on the sweeping angle (−α/2 to α/2). Exploiting the orthogonality between the blasting nozzle arm and direction of sweep, the strip width can be derived as:(7)$$S_{w}=2\left(\sqrt{R^{2}-B^{2}\sin\left(\frac{\alpha^{2}}{2}\right)}-D\cos\left(\frac{\alpha }{2}\right)\right)\sin\left(\frac{\alpha }{2}\right).$$
where $D=d_{n}$. So, the number of strips can be calculated as:(8)$$n_{s}=floor\left(\frac{W}{S_{w}}\right)+1.$$

The floor(.) function is used because, the number ($\frac{W}{S_{w}}$) may not be a positive integer (it may be a fraction). For this reason, the robot has to cover ($n_{s}-1$) number of strips of width $S_{w}$, and an additional strip of smaller width for complete area coverage. The time taken for the ($N_{p}-1$) rotational movement is calculated as:(9)$$T_{rotation}=\left(n_{s}-1\right)\left(\frac{\pi }{\omega_{l}}+\frac{S_{w}}{v_{l}}\right).$$

Further, given a strip of length, number of nozzle sweeps is given by
(10)$$n_{w}=\frac{L}{B}$$
where *B* should be chosen to get an integer $n_{w}$ as the number of nozzle sweeps cannot be a fraction. For the last strip, the sweep angle will be less then α as the width is less then $S_{w}$. Defining $\alpha_{s}$ to be the sweep angle, the time taken for sweeping the last strip can be computed to be $n_{w}\frac{\alpha_{s}}{\omega_{a}}$, where $\omega_{a}$ is the angular velocity of the nozzle arm. The total sweeping time and consequently the overall blasting time for stop-sweep strategy is derived as:(11)$$\begin{array}{c} T_{sweep}=\left(n_{s}-1\right)n_{w}\frac{\alpha }{\omega_{a}}+n_{w}\frac{\alpha_{s}}{\omega_{a}}\\ T_{blast}=\left(n_{s}-1\right)n_{w}\frac{\alpha }{\omega_{a}}+n_{w}\frac{\alpha_{s}}{\omega_{a}}+\left(n_{s}-1\right)\left(\frac{\pi }{\omega_{l}}+\frac{S_{w}}{v_{l}}\right)+n_{s}\frac{L}{v_{l}}. \end{array}$$

To find the sweep angle required for complete area coverage in minimum time, the following optimization problem needs to be solved, that is,
(12)$$\begin{array}{c} \;\;\;\;\;\;\;\min_{\alpha }T_{blast}\\ \mathrm{subject}\;\mathrm{to}\;\alpha_{min}\leq \alpha \leq \alpha_{max},\;\omega_{min}\leq \omega_{a}\leq \omega_{max}. \end{array}$$
where $\alpha_{min},\alpha_{max}$ are the minimum and maximum sweep angle possible by the nozzle; $\omega_{min},\omega_{max}$ are the minimum and maximum angular velocity of the blasting arm respectively. The optimization problem’s solution is not straightforward due to the involvement of constraints on α and $\omega_{a}$. Hence, the problem is converted to an unconstrained optimization problem by using the penalty function method.

For this purpose, the constraints are modified to inequality constraints as:(13)$$g\left(\alpha \right)=\left[\begin{array}{c} g_{1}\left(\alpha \right)\\ g_{2}\left(\alpha \right)\\ g_{3}\left(\dot{\alpha }\right)\\ g_{4}\left(\dot{\alpha }\right) \end{array}\right]=\left[\begin{array}{c} \alpha -\alpha_{max}\\ -(\alpha -\alpha_{min})\\ \dot{\alpha }-\omega_{max}\\ -(\dot{\alpha }-\omega_{min}) \end{array}\right]\leq 0.$$

The penalty function based unconstrained cost is formulated as:(14)$$\begin{array}{c} \min_{\alpha }T_{blast}^{p}\left(\mu \right)=\min_{\alpha }T_{blast}+\mu \left(k\right)P_{t}\left(\alpha \right)\\ \mathrm{where}\;P_{t}\left(\alpha \right)=\sum_{1}^{4}\left\{\max\left(0,g_{i}\left(\alpha \right)\right)\right\}\;\mathrm{and}\;\mu \left(k+1\right)=\mu \left(k\right)+1,\;\mu \left(0\right)\in R^{+}. \end{array}$$

The constant μ(k) is called the penalty factor, which is monotonically increasing with each sample. The function $P_{t}\left(\alpha \right)$ is the penalty function considered for our purpose. For one sweep, the energy spent by the blasting arm can be given by $\frac{1}{2}I_{a}\omega_{a}^{2}$, where $I_{a}$ is the moment of inertia of the nozzle arm.

For minimizing the sweep energy for the whole blasting area, one has to minimize
(15)$$\begin{array}{c} E_{sweep}=n_{s}n_{w}\frac{1}{2}I_{a}\omega_{a}^{2}\\ s.t\;(\omega_{min}\leq \omega_{a}\leq \omega_{max}). \end{array}$$

Note that, the constraint is already there in the minimum time problem (Equation 12). Hence, the overall unconstrained multi-objective optimization problem is given by
(16)$$\begin{array}{c} \min_{\alpha ,\omega_{a}}\left(T_{blast}^{p}\left(\mu \right),E_{sweep}\right)=\min_{\alpha ,\omega_{a}}\left(T_{blast}+\mu \left(k\right)P_{t}\left(\alpha \right),\frac{1}{2}I_{a}\omega_{a}^{2}\right)\\ \mathrm{where}\;P_{t}\left(\alpha \right)=\sum_{1}^{4}\left\{\max\left(0,g_{i}\left(\alpha \right)\right)\right\}\;\mathrm{and}\;\mu \left(k+1\right)=\mu \left(k\right)+1,\;\mu \left(0\right)\in R^{+}. \end{array}$$

**Remark** **1.**
*It is important to note that, the section derived the optimization problem, both in constrained ((Equation 12)–(Equation 15)) and unconstrained form (Equation 16). This is done intentionally, as many techniques need the optimization problems to be converted into the unconstrained form. So, it is up to the designer’s freedom in choosing a particular technique to solve the optimization problem.*


## 5. Result and Discussion

This section discusses the validation and simulations for the modeled problem. The line-sweep strategy uses a fixed robot footprint, hence there is no need to perform optimization in this strategy. In the case of the stop-sweep strategy, the robot footprint is varying with the movement of the arm. This arm movement is modeled mathematically and an expression (Equation (Equation 16)) is obtained for maintaining the optimal robot functional footprint. Solving the Equation (Equation 16) provides us the optimal values of α and $w_{a}$. As we can see from Equations (Equation 11) and (Equation 15), α and $w_{a}$ effect robot footprint, and obtaining the optimal value for both variables aids in minimizing the robot’s time and energy. We have incorporated a fast elitist genetic algorithm to solve this multi-objective optimization problem which is discussed further in this section. The genetic algorithm applied is validated by considering different scenarios for the robot operation and also different possible constraints to observe reasonable outputs.

### 5.1. Line-Sweep Strategy

As mentioned in Section 4.1, the Line-Sweep strategy is one of the two approaches to execute hydro-blasting on a given area of ship hull. The relation between total time taken for hydro-blasting ($T_{blast}$) and linear velocity ($v_{l}$) for fixed arm robot to cover areas of 10 × 10 and 20 × 20 is given in Figure 8. From the simulation, it is clear that the total time for hydro-blasting is inversely proportional to the robot’s linear velocity. A closer look at a $v_{l}$ vs. $T_{blast}$ curves for 10 × 10 and 20 × 20 areas we can understand that the linear velocity significantly affects efficiency in terms of time irrespective of the coverage area. Hence, the best strategy for time-efficient hydro-blasting is to operate the robot at the maximum permissible linear velocity.

For the Line-Sweep Strategy, the robot’s functional footprint is a constraint within the nozzle diameter which encourages us to explore the Stop-Sweep Strategy with a functional footprint that is not limited to the nozzle diameter but also variable in terms of sweep angle and rate of sweeping. The simulation and performance analysis for the Stop-Sweep strategy is given in the next sub-section.

### 5.2. Stop-Sweep Strategy

For the case of the Stop-Sweep cleaning strategy, the total time for hydro-blasting depends upon more than one factor including sweep angle α and $w_{a}$ sweep. Hence the functional footprint of the for this robot is defined by the α and $w_{a}$. The functional footprint for the hydro-blasting robot must maximize the efficiency of the hydro-blasting operation. In Stop-Sweep strategy, the efficiency in area coverage is determined by minimal energy (Equation (Equation 15)) and minimal time (Equation (Equation 11)). The solution to this problem can be arrived by performing a constraint multi-objective optimization. A Genetic Algorithm is a useful tool for realizing multi-objective optimization problems. Regardless of the complexity in the objective functions and constraints, Genetic Algorithm converges to the optimal solution as it performs sequences of iterations. The genetic algorithm is inherently parallel for its nature of computation, and it is well known for its reliability [25,26].

### 5.3. NSGA-II

Genetic algorithms (GA) are a class of evolutionary algorithms. GA is meta-heuristic algorithms which are highly suitable for complex optimization problems [27,28]. The general GA are customized to optimize more than a single objective to address the multi-objective problems. The single objective optimization problems result in a single optimal problem; however, the multi-objective optimization problem provides a group of points, also known as Pareto-Optimal solution set. Pareto-front contains the dominant solution under two conflicting objectives. To achieve the Pareto-Optimal solution, there exists different Multi-Objective Evolutionary algorithms (MOEA) including NSGA [29], NSGA-II [30], MOEA/D [31], PESA-II [32] and so forth. However, the NSGA-II algorithm is known for its simplicity of implementation and effectiveness. NSGA-II algorithm is an improved version of NSGA using the fast non-dominated sorting concept [30]. The steps for NSGA-II explained in Figure 9.

The step by step procedure in Figure 9 the simple flow process of the algorithm. The algorithm starts with a set of a population $P_{o}$ of size *N*. Off-springs are generated from this set of the population with possible crossovers and mutations. Then the off-spring population and the parent population are combined to form the population $R_{t}$ of population $P_{t+1}$ with size 2N which ensures elitism population is included. The population is sent through the fast-non-dominating-sort to obtain the set *F*. The principal elements of this are chosen as it is sorted in descending order based on the crowded distance, making sure that only eligible elements remain in the population. A new population set $P_{\left(t+1\right)}$ is generated from these selected elements, and the iteration continues till the generation limit is reached.

Recently, the NSGA-II algorithm has been used extensively in the area of NN optimization [33], Design optimization [34,35], Network optimization [36] and so forth. Considering the strong precedence and use cases, we use NSGA-II as a tool for optimizing our multi-objective problem in this research work. A toolbox provided by [37] called pymoo for Python is used to implement NSGA-II for the problem. The toolbox gives access to choose the different parameters for the algorithm. We have experimented with the algorithm by altering various parameters and have validated our experimental results which are discussed in the following section.

### 5.4. Validation of Results

The optimization is carried out in multiple ways. We have simulated the problem to arrive at best suitable candidate for functional footprint for the robot. The effect of cost function parameters, effect of algorithm parameters and effect of constraints on the optimal solution is identified thoroughly. The simulation is done with the help of pymoo Python module and MATLAB is used to plot the data.

#### 5.4.1. Effect of Population

The optimization has been done for three different population sizes-50, 500, 1000. In all the three cases, the number of generations for the algorithm is fixed as 100. The coverage area has been fixed as 10 × 10 m, angular velocity of the arm and maximum sweeping angle are fixed at 1 rad/s and pi/3 respectively. For the population size of 50, we could observe less number of points in the solution space Figure 10 ($x_{1}$ represents α, the sweep angle, and $x_{2}$ represents the $w_{a}$, the angular velocity).

An increase in population gives more solutions on the Pareto-set and design space. This gives more confidence to choose the optimal set of values. Figure 11a,b shows the simulation outcome initializing the algorithm with an initial population of 500 for 10 × 10 area. ($x_{1}$ represents α, the sweep angle, and $x_{2}$ represents the $w_{a}$, the angular velocity).

Increasing the initial population to 1000 gives more number of $x_{1}$ and $x_{2}$. Figure 12 shows the outcome of optimization where the algorithm is initialised with a population of 1000.

#### 5.4.2. Effect of Area as a Cost Function Parameter

Simulation has been carried out for three different areas: 5 × 5, 10 × 10 and 20 × 20. In the above mentioned case, the number of generations for Algorithm is fixed as 100.

The upper bound of maximum sweeping angle and angular velocity are fixed at *pi*/6 and 1 rad/sec, respectively. For area 5 × 5, we could observe a lower number of points in the solution space Figure 13, compared to area 10 × 10 Figure 14 and 20 × 20 Figure 15 ($x_{1}$ represents α the angle of arm rotation and $x_{2}$ represents the angular velocity).

In each case, the linear velocity and angular velocity of the robot is fixed as 0.1 m/s and 0.1 rad/s, respectively. The best α and $w_{a}$ has been chosen from the Pareto-front. However, in this instance, we choose the α from the solution set which is less dominant over both objectives such that the chosen value is not dominant towards any objective. The most number of optimal solution for α was found in an interval of [0.48, 0.52]. In the case of optimal solution for $w_{a}$ falls in the interval of [0.1, 0.3]. The above-mentioned behaviour in obtaining solution can be more significantly observed on simulations with a population size of 500 and 1000.

#### 5.4.3. Effect of Constraint on Cost Function Parameter

A set of simulations were carried out for analysing the effect of the constraint equation on reaching the optimal solution. The simulation was run with the algorithm initialized for a population size of 1000 and an area of 10 × 10. The constraint for $w_{a}$ has been kept fixed greater than 0.4 rad/s but less than 1 rad/s. The constraint on α has been varied from 1, <pi/3, 2, <pi/4 and <pi/6. The result of the simulation is given in Figure 16, Figure 17 and Figure 18 respectively. For α < pi/6 gives less number of optimal solution set where as α < pi/3 gives most number of optimal solution. In the three simulation output, we observed more number of feasible solution points distributed on α in the interval of [0.45, 0.52]. This indicates, for this particular scenario, the changes in constraints have a minimal impact on the algorithm to find out the optimal solution. However, the algorithm is not able to find more number of solutions while constraint on α is less than pi/6 rad. The simulation results shows that the Pareto-optimal solutions for α is within an interval of [0.45 rad, 0.52 rad] and the best solution for $w_{a}$ is with in an interval of [0.1, 0.5]. We choose the center of the Pareto solutions, that is, α = 0.48 rad and $w_{a}$ = 0.3 rad/s as the best solution for completing the hydro-blasting task.

#### 5.4.4. Experiments

The experiments were carried out on the physical robot to validate the findings from the simulations. The robot is operated on a vertical metal wall that replicates the conditions of a ship hull. The robot is powered using tethered communication and power cable. The power to the robot is supplied by a digital power supply. The voltage and current consumption is recorded using the ammeter and voltmeter on the power supply unit. The experiment is carried out for 5 m × 5 m area. There are four sets of angular velocity and sweep angle that are chosen for the robot. The time taken for completion of hydro-blasting has been recorded in all experiments using a stop watch. The area covered by the robot is estimated by tracking the movement of robot arm. First experiment is carried out with angular velocity equal to 0.5 rad/s and sweep angle equal to $55^{\circ }$ (Figure 19). Second set of experiments are carried out with angular velocity equal to 0.4 rad/s and sweep angle equal to $41^{\circ }$ (Figure 20). Similarly the third set of experiments is performed with the optimal footprints (angular velocity and sweep angle equal to $28^{\circ }$ and 0.3 rad/s respectively are show in Figure 21). Final set of experiment, is conducted with $25^{\circ }$ and 0.2 rad/s (Figure 22). The outcome of the experiment is tabulated in the Table 1. Further, the results for line sweep strategy is given in the Table 2.

*Comparison between LS and SS Strategy:* It is already mentioned that, the fixed arm robot can not follow the SS strategy (Arm cannot rotate). The Table 2 presents the time and energy consumed by the two types of Hornbill robots, while following a LS strategy. The time taken for the LS strategy is more than the SS strategy for both robots. This is expected since, the SS strategy for the moving arm robot can cover strips of larger width, and therefore the area is covered faster. However, it is interesting to note that, the energy consumption in LS strategy is always less compared to the energy consumption for the SS strategy. This can be explained by the additional energy need for moving the arm in SS strategy.

From the results presented in the table, it can be observed that the optimal case (third experiment) leads to the consumption of minimum time and energy, which confirms our claim.

## 6. Conclusions

This paper calculates the optimal time and energy consumption for a ship hull maintenance robot, Hornbill for a given scenario. Two coverage strategies named as Line Sweep and Stop Sweep are proposed for two types of hydro-blasting robot Hornbill. The derived expressions for the time and energy spent during blasting are depends on the sweeping angle and speed of the nozzle arm. Multi-objective optimization is performed to optimize the time and energy along with the complete coverage. From the multiple simulations and validation experiments, the multi-objective GA is found to be efficient at computing an optimal solution for different applications. With these optimal values, ship-hull maintenance can be performed with much more efficiency. Future work will focus on the dynamic optimization for the hydro blasting robot Hornbill, inclusion of criteria for uniform blasting and cleaning and also the effect of obstacles and uneven surface in robot performances.

## Figures and Tables

**Figure 1 sensors-21-01194-f001:**
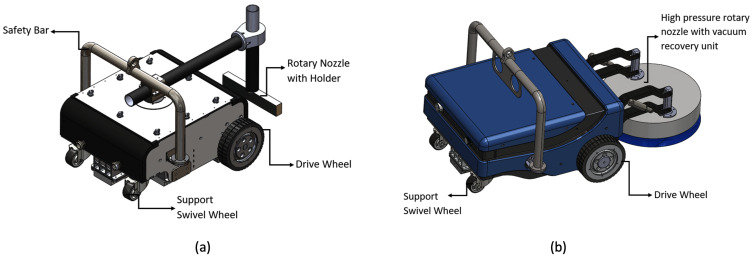
Robot Hornbill: (**a**) isometric view of open arm robot, (**b**) view of enclosed arm robot.

**Figure 2 sensors-21-01194-f002:**
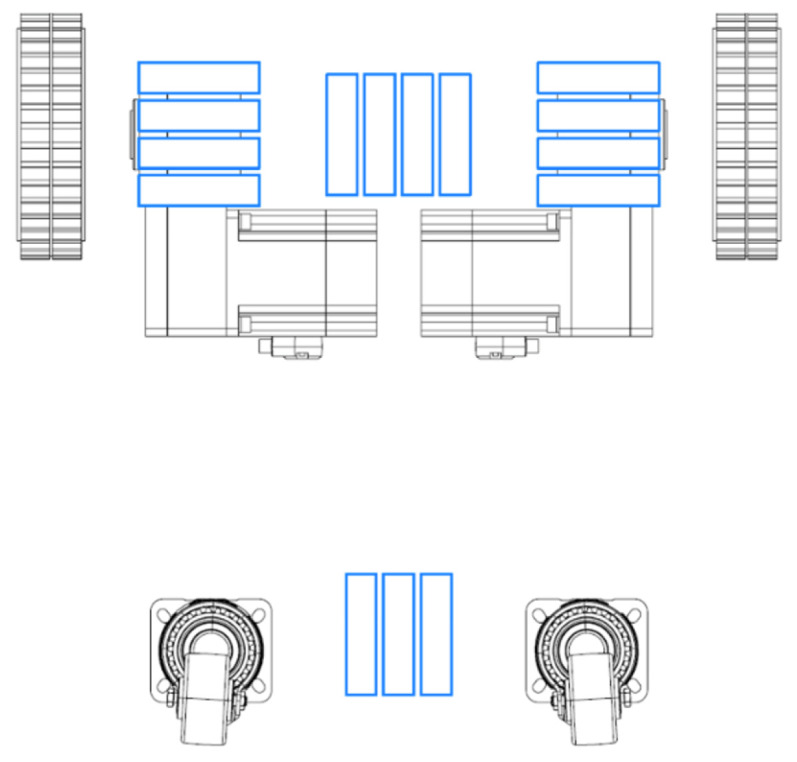
Magnets Placement.

**Figure 3 sensors-21-01194-f003:**
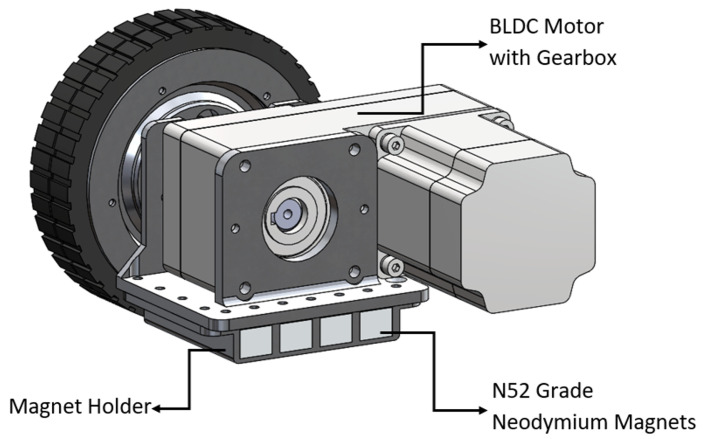
Locomotion Unit with Magnets.

**Figure 4 sensors-21-01194-f004:**
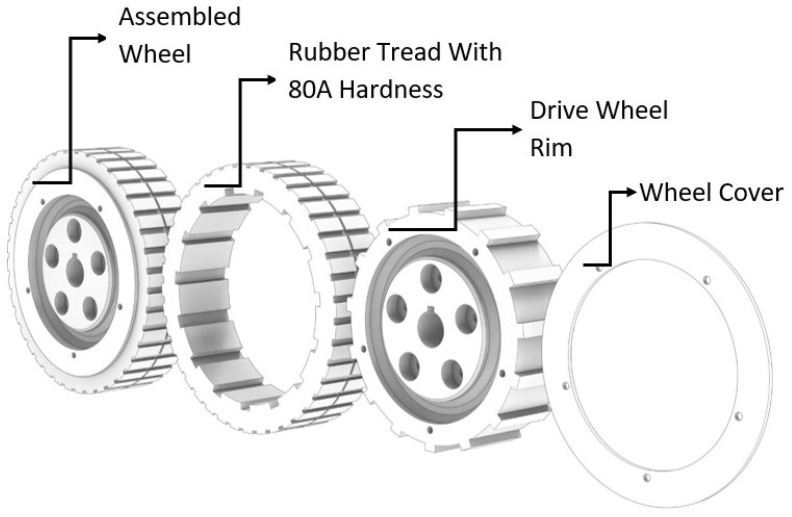
Drive Wheel Structure.

**Figure 5 sensors-21-01194-f005:**
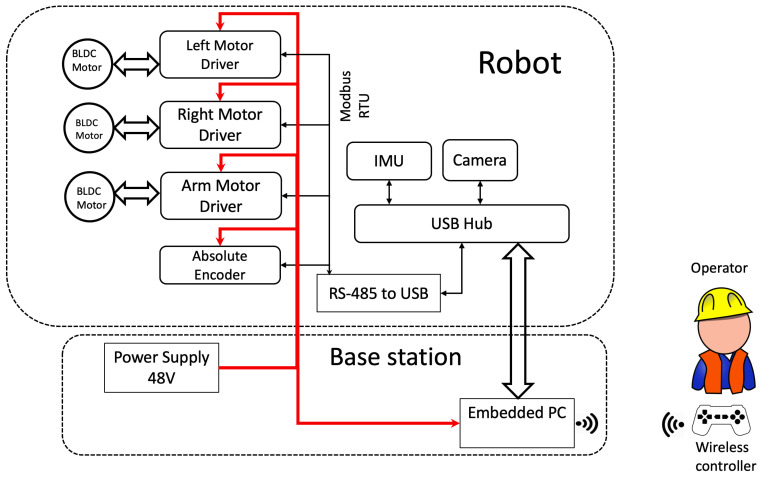
Flow Diagram of the Control System.

**Figure 6 sensors-21-01194-f006:**
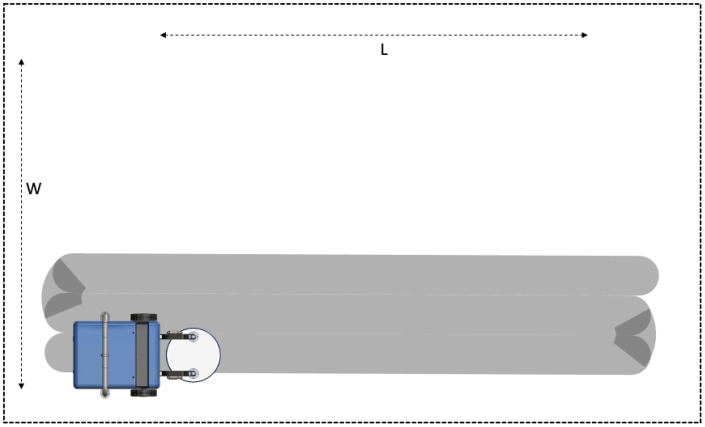
Line-Sweep Strategy.

**Figure 7 sensors-21-01194-f007:**
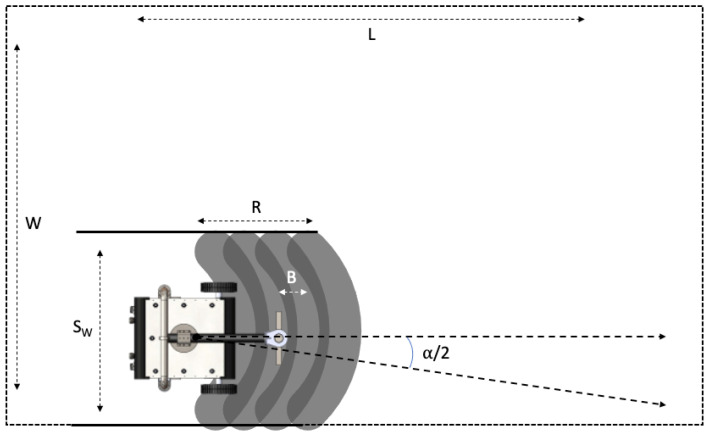
Stop-Sweep Strategy.

**Figure 8 sensors-21-01194-f008:**
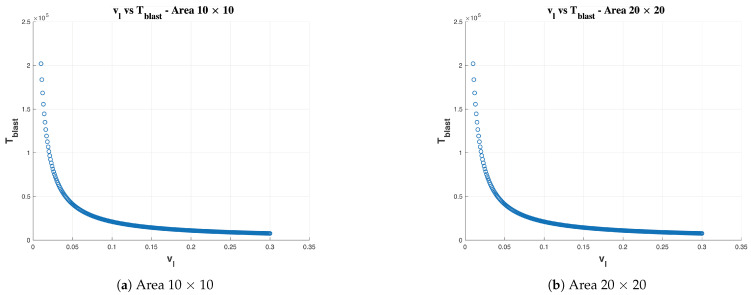
Time taken for basting ($T_{blast}$) vs. linear velocity $v_{l}$ in line sweep strategy.

**Figure 9 sensors-21-01194-f009:**
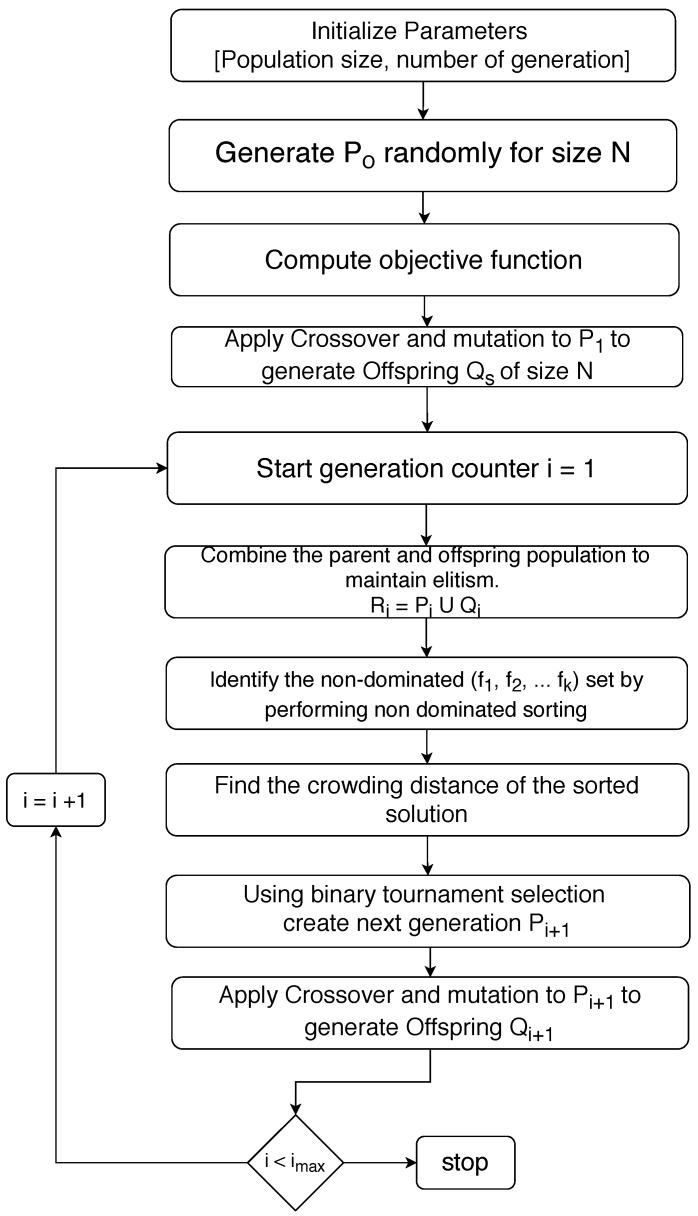
Procedure for NSGA-II algorithm implementation.

**Figure 10 sensors-21-01194-f010:**
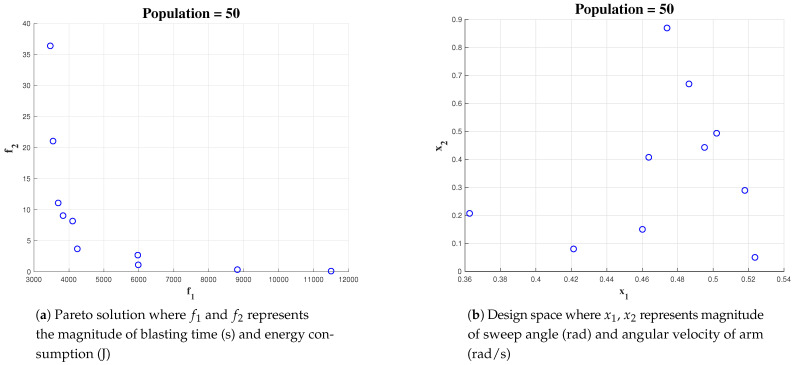
Simulation result for Population of 50 with area 10 × 10.

**Figure 11 sensors-21-01194-f011:**
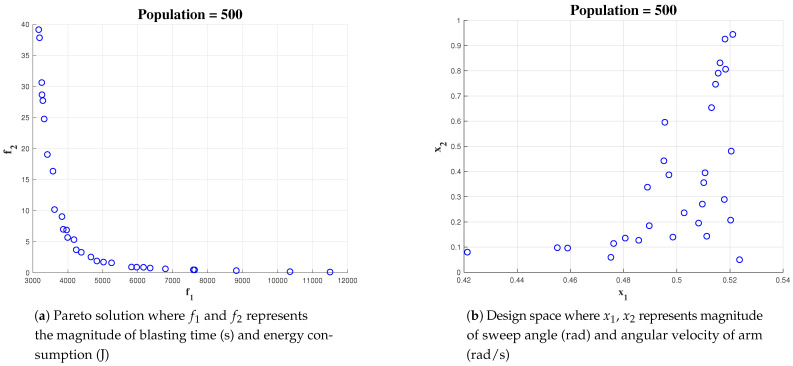
Simulation result for Population of 500 with area 10 × 10.

**Figure 12 sensors-21-01194-f012:**
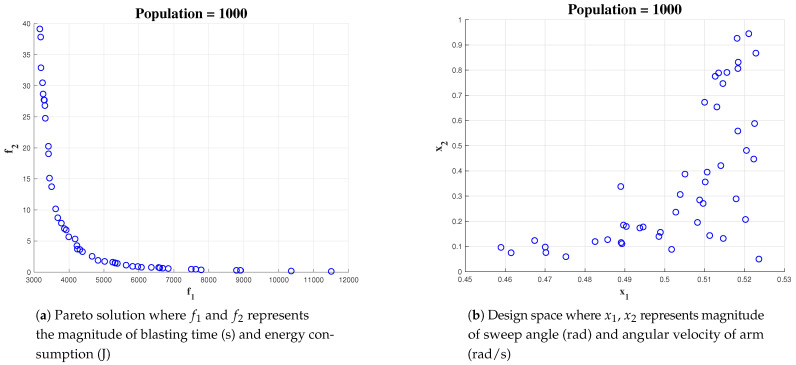
Simulation result for Population of 1000 with area 10 × 10.

**Figure 13 sensors-21-01194-f013:**
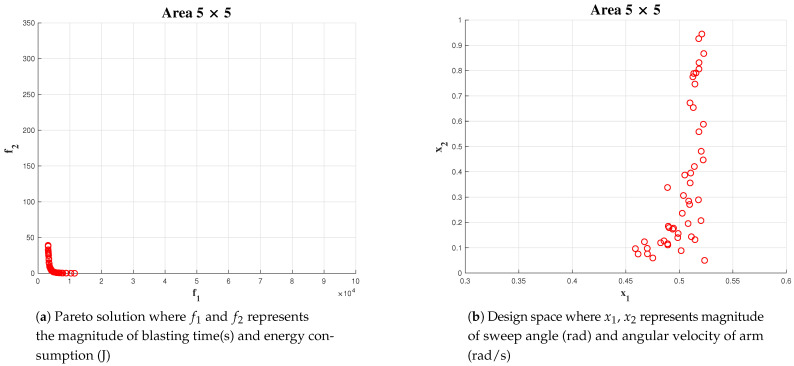
Simulation result for area 5 × 5 population of 50.

**Figure 14 sensors-21-01194-f014:**
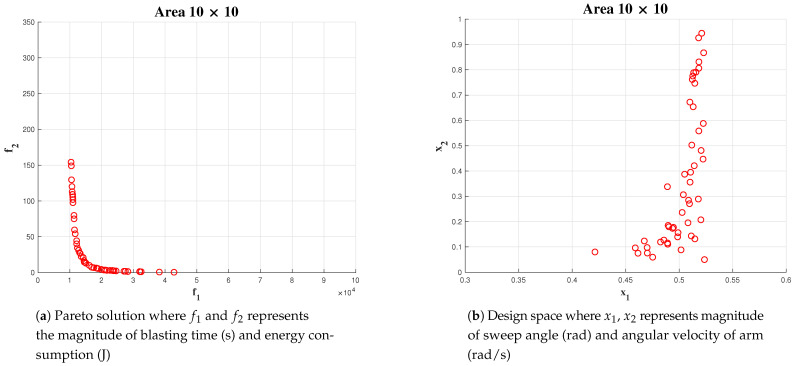
Simulation result for area 10 × 10 population of 50.

**Figure 15 sensors-21-01194-f015:**
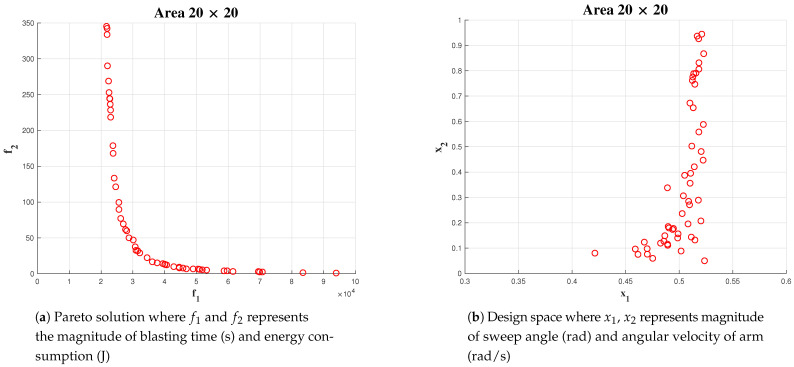
Simulation result for area 20 × 20 population of 50.

**Figure 16 sensors-21-01194-f016:**
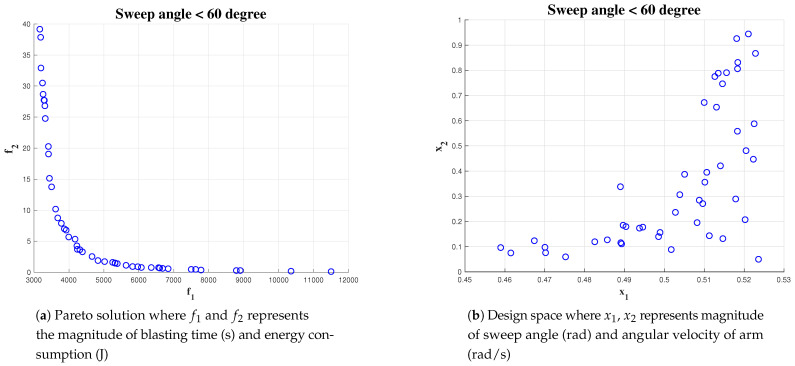
Simulation result for α<60, area 10 × 10 and population of 1000.

**Figure 17 sensors-21-01194-f017:**
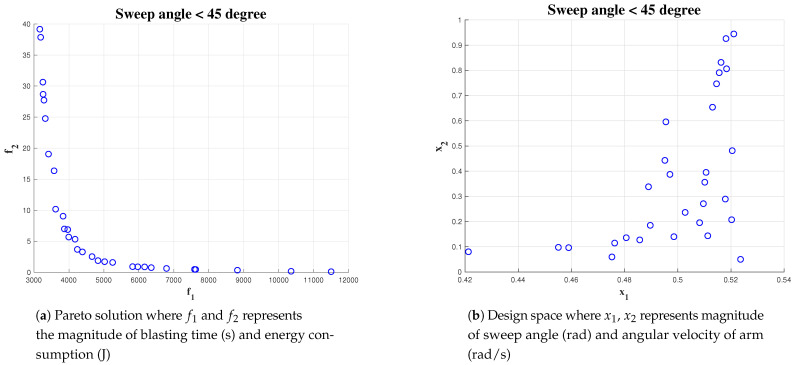
Simulation result for α<45, area 10 × 10 and population of 1000.

**Figure 18 sensors-21-01194-f018:**
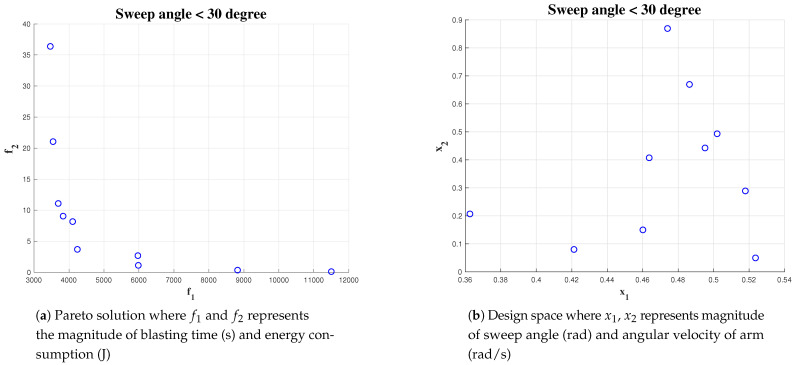
Simulation result for α<30, area 10 × 10 and a population of 1000.

**Figure 19 sensors-21-01194-f019:**
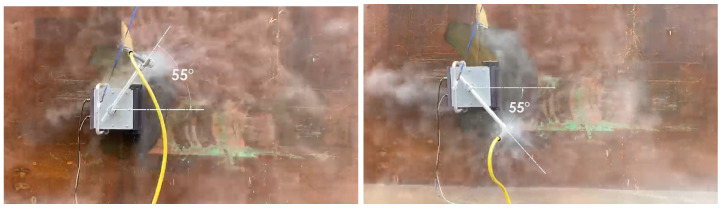
Experiment conducted for $55^{\circ }$ sweep angle.

**Figure 20 sensors-21-01194-f020:**
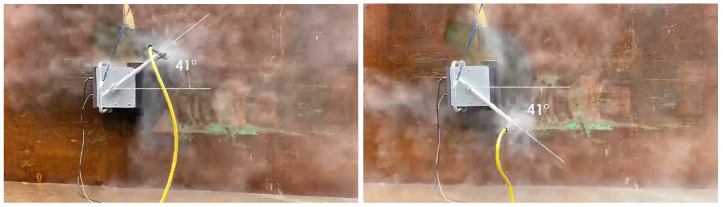
Experiment conducted for $41^{\circ }$ sweep angle.

**Figure 21 sensors-21-01194-f021:**
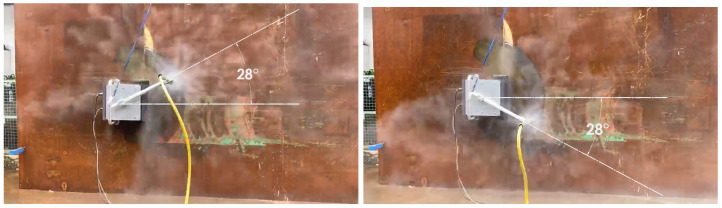
Experiment conducted for $28^{\circ }$ sweep angle.

**Figure 22 sensors-21-01194-f022:**
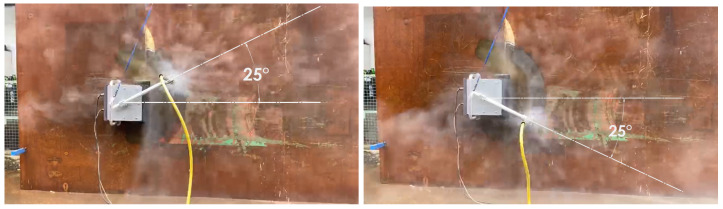
Experiment conducted for $25^{\circ }$ sweep angle.

**Table 1 sensors-21-01194-t001:** Experiment results for Stop-Sweep strategy.

% Area Covered	Voltage (V)	Avg Current (A)	Total Time (s)	Sweep Angle (Deg)	Arm Velocity (rad/s)	Energy (J)
99.2%	48.02	3.32	11,218	55	0.5	159.42
98.9%	48.11	3.26	11,314	41	0.4	156.83
99.2%	48.23	3.24	11,207	28	0.3	156.27
99.1%	48.17	3.21	11,229	25	0.2	154.63

**Table 2 sensors-21-01194-t002:** Experiment results for Line-Sweep strategy.

Platform Type	% Area Covered	Voltage (V)	Avg Current (A)	Total Time (s)	Energy (J)
Fixed Arm	99.1%	48.10	3.01	11,229	144.78
Movable Arm	98.5%	48.01	3.25	15,312	156.21

## Data Availability

Not applicable.

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
