# Peer review of "An Optimal Footprint Based Coverage Planning for Hydro Blasting Robots"

_sensors, 2021, doi:10.3390/s21041194_

Round 1
Reviewer 1 Report
The authors discuss a coverage planning for hydro blasting robots on hulls. The problem is well described, and the authors are thoroughly describing the state-of-the-art. However, the impression of the reader may be at the end, that there is an unbalance between the line-sweep and stop-sweep strategies. The line-sweep is a classical problem, that has roots to CNC machining, while the stop-sweep is slightly an artificially composed problem. The authors do not answer the main question, if any of these to strategies is superior or not. The authors also lack to address the question of how coverage is calculated (I suppose this not only depending on where the head position is located, but also that how much rust/contamination should be removed).
My comments and remarks to the authors:
- Section 2. Hornbill Architecture should have a reference to [1]. Many figures are from that source.
- Comparison of the results to [2].
- What does area mean in Table 1?
- Compare LS to SS like in Table 1.
- Define criteria for cleaning in GA.
[1] V. Prabakaran et al., "Hornbill: A Self-Evaluating Hydro-Blasting Reconfigurable Robot for Ship Hull Maintenance," in IEEE Access, vol. 8, pp. 193790-193800, 2020, doi: 10.1109/ACCESS.2020.3033290.
[2] Anh Vu Le, Phone Thiha Kyaw, Prabakaran Veerajagadheswar, M.A. Viraj J. Muthugala, Mohan Rajesh Elara, Madhu Kumar, Nguyen Huu Khanh Nhan, Reinforcement learning-based optimal complete water-blasting for autonomous ship hull corrosion cleaning system, Ocean Engineering, 2020, 108477, ISSN 0029-8018, https://doi.org/10.1016/j.oceaneng.2020.108477.
Author Response
Respected reviewer,
We sincerely thank you for your time and valuable comments in reviewing our paper, I greatly
appreciate it. We have revised the paper taking into account the reviewer comments seriously.
We are now submitting the revised manuscript for review and publication. The changes in the
manuscript has been highlighted in cyan color.
Sincerely,
Thejus Pathmakumar

Reviewer 2 Report
Article in subject of Sensors Journal.
The paper is difficult to read. Article consist many bugs need improve.
Some my observations I present using many comments. The are prepared mainly in order of reading
General comment
In article I found many mistakes. The main problem is in description of figures. Need improve this problem to improve an article read ability. My advice autors can find in detailed prepared comments.
Comment 1
You wrote
Figure 1.(a) Isometric View of Open Arm Hornbill, (b)Isometric View of Enclosed Hornbill
If author agree I propose use of main description of Figure 1 and next to describe of (a) and (b)
For example
Figure 1. Robot Hornbill: (a) isometric view of open arm, (b) view of enclosed robot
I propose also to increase of font of text for description of robots parts
Comment 2
Figure 3. I propose increase font size
Comment 3
122 line
For the fixed arm Hornbill robot, the blasting nozzles are attached to two fixed arms. In such a case, a lawn-mower type area coverage is suitable, which is termed as line-sweep (LS) strategy (c.f. Fig.6).
i propose use Figure 6
For the fixed arm Hornbill robot, the blasting nozzles are attached to two fixed arms. In such a case, a lawn-mower type area coverage is suitable, which is termed as line-sweep (LS) strategy (c.f. Figure.6).
Comment 4
Please number of of relation in 6 page.
After relation I propose descript of parameter
ns, W, fr
I propose use of description of relation in other form
ns – int(W/fr)+1
You no need to describe of function “floor”
Comment 5
A simple but effective coverage strategy will be to stop at fixed lengths (B); sweep the nozzle through a predefined angle (−α2−α2); and repeat the process till the complete length L is covered (c.f.Fig 7).
(−α2−α2) is it realy true ? I suppose that it is (−α2, α2)
Comment 6
The time taken for an LS strategy depends on the robot velocities. Similarly, the time taken for the blasting and the required energy demand varies with different sweeping angles, even though the sweeping angle is kept within 600.
Sweeping angle please use also meaning
Comment 7
Please renumber of relation
Tblast=Tsweep+Ttranslation+Trotation (1)
Comment 8
You wrote
In this strategy, the robot moves without the nozzle rotations. So, the sweep time is equal to the time required for translations. Hence,
Tswee p=Ttranslation=nsLvl
Where n
s is the number of strips along the total area,vlis the linear velocity of the robot.
The number of strips required for complete area coverage can be derived as:
ns=W/dn
where dn is the diameter of the nozzle.
Please number of relation in order
Also please improve of pdf file… I see in all relations -- but when I copy I get =
Please check all relations in paper and number all !!
Comment 8
Please increase some of figure 8. Please descript of left and right graphs as a) and b) and write in Figure 8 description
Figure 8.Time taken for hydro basting in line sweep strategy: a)….., b)……..
Comment 9
For the case of Stop-Sweep cleaning strategy, the total time for hydro-blasting depends upon more than one factor including sweep angle alpha and wa sweep. Hence the functional footprint of the for this robot is defined by the alpha and wa.
In article and Figure 7 you not use alpha but a. Please check it and improve in
all article
Comment 10
Please consider to present Procedure NSGA II using schemat algorithm
Comment 11
You wrote
Figure 9.Pareto solution and Design Space for for a population of 50
Please describe the left and right graphs as a) and b)
Figure 9.Pareto solution and Design Space for for a population of 50: a)…., b)…..
That same Figure 10, 11, 12, 13, 14 and Figure 15-17
Comment 12
Line 267
Increasing of initial population to 1000 gives more number ofx1andx2. Figure [] shows the outcome of optimization where the algorithm is initialised with a population of 1000.
Please improve number of cited figure
Comment 13
Line 259
The area has been fixed as10x10meters, angular velocity of the arm and maximum sweeping angle are fixed atpi/3 rad 1 rad/sec respectively.
Order was false
The area has been fixed as10x10meters, angular velocity of the arm and maximum sweeping angle are fixed at 1 rad/sec pi/3 rad respectively.
Comment 14
You present graphs Figure 9 -11. You show results in different space
(min and max value of x1 x2). In Figure 11 you wrote in design space graphs coordinations f1 f2 ????
Comment 15
Figures 12-14 has the same description
Figure 12.Pareto solution and Design Space for for a population of 50
Figure 13.Pareto solution and Design Space for for a population of 50
Figure 14.Pareto solution and Design Space for for a population of 50
You need show in description of figure the difference of figures (an area)
That same for figures 15-17
Figure 15.Pareto front and Design Space for for a population of 1000
Figure 16., Objective Space and Design Space for for a population of 1000 ????
Figure 17.Pareto front and Design Space for for a population of 1000
You need show in description of Figure the difference of figures (sweep angle)
You make many mistakes in this descriptions
Comment 16
General advice for description for all figures from analysis
You present double graphs in Figures 9-14 descripted that same as
Figure xx.Pareto Solution and Design Space for a population of YYYY
You not presented which graphs is Pareto which is Design Space
It need to show as a) and b)
In description you not write for which parameter you present graph
You don't show units, that is, they are relative values
For Figure 15 and 17 you wrote Pareto Front and Design Space for a population …..
Is it difference it is other parameter as for previous figures
But for Figure 16
Figure 16.Objective Space and Design Space for for a population of 1000
Is it different from Figure 15 and 17
Comment 17
I propose add nomenclature to article
Author Response

(The authors gave the same response as above.)

Round 2
Reviewer 1 Report
The authors have replied to all answers from the reviewers. The quality and presentation of the article has been increased.
Author Response
Respected reviewer,
We sincerely thank you for your time and valuable comments in reviewing our
paper, I greatly appreciate it. We have revised the paper taking into account
the reviewer comments seriously. We are now submitting the revised
manuscript for review and publication. The changes in the manuscript has
been highlighted in green color.
Sincerely,
Thejus Pathmakumar

Reviewer 2 Report
Article was improved. I have only some editorial advice.
Comment 1
You wrote
"....... achieve the Pareto-Optimal solution, there225exists different Multi-Objective Evolutionary algorithms (MOEA) including NSGA[29], NSGA-II[30],226MOEA/D[31], PESA-II[32] etc. However, the NSGA-II algorithm is known for its simplicity of227implementation and effectiveness. NSGA-II algorithm is an improved version of NSGA using the fast228non-dominated sorting concept [30]. The steps for NSGA-II explained below."
Please add space before [ ]
NSGA [29], NSGA-II [30],226MOEA/D [31], PESA-II [32] etc.
Please check it in all paper
and Figure 9 need citation in text
The steps for NSGA-II explained in Figure 9.
Comment 2
You wrote
Author Contributions: Conceptualization, Thejus Pathmakumar. and Madan Mohan Rayguru.; methodology,
Sriharsha Ghanta and Madan Mohan Rayguru.; software, Thejus Pathmakumar and Sriharsha Ghanta.; validation,
Manivannan Kalimuthu, Madan Mohan Rayguru. and Rajesh Mohan Elara.; analysis, Thejus Pathmakumar.
Madan Mohan Rayguru; Manivannan Kalimuthu and Rajesh Mohan Elara; Original draft, Thejus Pathmakumar,
Madan Mohan Rayguru, Sriharsha Ghanta, Manivannan Kalimuthu, and Mohan Rajesh Elara
Please use short
Thejus Pathmakumar T.P.
please add it to list authors of article
Engineering Product Development Pillar, Singapore University of Technology and Design, Singapore;
pathmakumar_thejus@mymail.sutd.edu.sg, (T.P.); madan_rayguru@sutd.edu.sg, (M.R.);
Comment 3
Please check spaces after dot "." in paper
Please check spaces after comma "," in paper
Author Response
Respected reviewer,
We sincerely thank you for your time and valuable comments in reviewing our
paper, I greatly appreciate it. We have revised the paper taking into account
the reviewer comments seriously. We are now submitting the revised
manuscript for review and publication. The changes in the manuscript has
been highlighted in green color (line numbers ​ 21, 29, 63, 95, 117, 180, 217,
220, 265, 271, 284, 287, 298, 301 and 308​ )
Sincerely,
Thejus Pathmakumar
